# Soil organic carbon fraction accumulation and bacterial characteristics in curtilage soil: Effects of land conversion and land use

Qingqing Cao[1‡], Bing Liu[2‡], Jinhang Wu[1], Xu Zhang[1], Wen Ma[3], Dongxu Cui[1]*

1 School of Architecture and Urban Planning, Shandong Jianzhu University, Jinan, China, 2 School of Municipal and Environmental Engineering, Shandong Jianzhu University, Jinan, China, 3 College of Chemistry and Chemical Engineering, Qilu Normal University, Jinan, China

‡ QC and BL are first authors on this work.
* cdx0109@sdjzu.edu.cn

**Data Availability Statement:** All bacterial files are available from the NCBI database (BioProject ID of PRJNA878090).

## Abstract

Conversion of curtilage land into cropland or grassland can have substantial effects on soil nutrition and microbial activities; however, these effects remain ambiguous. This is the first study to compare the soil organic carbon (SOC) fractions and bacterial communities in rural curtilage, converted cropland, and grassland compared with cropland and grassland. This study determined the light fraction (LF) and heavy fraction (HF) of organic carbon (OC), dissolved organic carbon (DOC), microbial biomass carbon (MBC), and the microbial community structure by conducting a high-throughput analysis. Curtilage soil had significantly lower OC content, the DOC, MBC, LFOC and HFOC of grassland and cropland soils were 104.11%, 55.58%, 264.17%, and 51.04% higher than curtilage soil averagely. Cropland showed notably high bacterial richness and diversity, with Proteobacteria (35.18%), Actinobacteria (31.48%), and Chloroflexi (17.39%) predominating in cropland, grassland, and curtilage soil, respectively. Moreover, DOC and LFOC contents of converted cropland and grassland soils were 47.17% and 148.65% higher than curtilage soil while MBC content was 46.24% lower than curtilage soil averagely. Land conversion showed more significant effects on microbial composition than land-use differences. The abundant Actinobacteria and Micrococcaceae population and the low MBC contents indicated a "hungry" bacterial state in the converted soil, whereas the high MBC content, Acidobacteria proportion, and relative abundance of functional genes in the fatty acid and lipid biosynthesis indicated a "fat" bacterial state in cropland. This study contributes to the improvement of soil fertility and the comprehension and efficient use of curtilage soil.

## Introduction

With the rapid urbanization and development of agricultural cultivation technology, modern agriculture and smart agriculture emerge and rapidly expand, which have become the main strategies of rural revitalization in China [1]. It indicates that numerous village settlements are

**Funding:** This study is supported by the National Key Research and Development Plan "Research on the spatial optimization and layout of rural and communities" (NO. 2019YFD1100801). Prof. Dongxu Cui is the program director. Natural Science Foundation of Shandong Province, China (ZR2020QC041), Qingqing Cao is the program director.

**Competing interests:** The authors have declared that no competing interests exist.

transforming into large communities, and the corresponding rural curtilage land is mostly being converted into cropland or grassland [2]. Preliminary visits and investigations have revealed that the average crop yield in the common cropland of northern China is 15742 kg/ha, but only 9895 kg/ha in curtilage-converted cropland. Moreover, farmers have reported that fertilization could not increase crop yield significantly [3]. Although the detailed effect of rural curtilage conversion on crop yield is unknown, it appears to have far-reaching consequences on soil fertility, nutrition and physicochemical properties [4]. Concretely, for soil physical structure, rural curtilage land soil (CS) shows relatively high soil pH value and soil hardness, which is unfavourable for nutrient retention [2, 5]. For soil biochemical property, CS has few constant input and accumulation of organic matter, and the microbial composition and activities may be notably different with other land types [6]. Fu et al. [7] shows that exogenous organic matter into constructed land soil can be quickly decomposed and consumption, the grain yield overground is relatively low. Tardy et al. [8] indicates that the microbial diversity is a good predictor of the land use transformation, which can directly influence the soil functioning, carbon cycling and plant growth above-ground. According to previous reports, we speculate that microbial composition and community, and soil organic matter (SOM), as key indicators of soil quality, should have significant differences among CS, curtilage-converted soil and agricultural land, they may also vital reasons for the low crop yield of curtilage-converted soil [5, 8]. However, the relevant research of SOM and microbial communities mostly focus on forest, grassland, wetland and cultivated soil worldwide, rather than curtilage land soil (CS) or curtilage-converted soils.

Sugar, organic acids, cellulose, hemicellulose, lignin, lipids, proteins, among, others are the primary components of SOM, which is a type of complex and relatively stable polymer organic compound formed by microbiological decomposition [9, 10]. Moreover, it is the nutrient reservoir that profoundly affects the soil physical, chemical, and biological properties [9, 11]. A previous study indicated that 1% of the organic carbon (OC) in the soil is equivalent to 270 kg/ha of nutrients [12]. Other studies have shown that lowering the organic matter content from 2% to 1.5% reduces soil fertility by 14% [13]. Consequently, it is crucial to investigate the organic components of CS. Microorganisms, animals, plants, and anthropogenic activities are the primary sources of SOM [14], which include returned crop residues such as roots and straw, along with human and animal manure, garbage, and sewage [15, 16]. However, increasing SOM alone cannot directly increase soil fertility; rather, the microbial decomposition, biosynthesis, and metabolism process is the most important link in the SOM transformation and reuse by crops [17, 18].

OM morphology and classification vary depending on the diversity and complexity of its potential sources [10]. Most OM added to soil is decomposed into carbon dioxide and water by microorganisms [19]. The undecomposed or partially decomposed residues of animals, plants, and microorganisms that can persist in soil are referred to as light fraction organic matter (LFOM), accounting for ~1–5% of OM [20]. Soil microorganisms account for only ~ 3–8% of SOM [21]. Small molecules of polysaccharides and organic acids are transferred between plants, microorganisms, and soil as dissolved organic matter (DOM) [15]. DOM typically accounts for ~3–8% of SOM, and contributes to microbial synthesis and metabolism [22]. There is a direct relationship between the carbon and nitrogen content of the abovementioned OM fractions and soil microbial activities. In addition, heavy fraction organic matter (HFOM) is the primary component of the SOM reservoir pool, accounting for over 60% of the remaining OM; it has a complex chemical structure and stable physicochemical properties [23]. Thus, SOM fraction characteristics and microbial activity can vary considerably based on land use or land conversion, affecting soil fertility and nutrition.

Emergence and augment of rural communities that transformed from courtyards are necessary and vital stages for the development of several countries. The biochemical characterization and change of CS should be concerned, as well as the relevant and effective improvement measures. With the support of the China National Key R&D Program of "Research on the spatial optimization and layout of rural and communities", we conducted this study. We takes curtilage land soil, curtilage-converted cropland soil (CCS) and curtilage-converted grassland soil (CGS) as research subjects, with soil of cropland and grassland set as control. This study aimed to: (1) present the composition and distribution of SOM fractions, (2) illustrate the bacterial composition and functional differences, (3) analyze the inter-connections between OM fractions and bacterial communities, and (4) describe the carbon and nitrogen accumulation potential in the five land types. This study contributes to the raise of effective measures for soil fertility advancement by improving our understanding of rural curtilage soils.

## Materials and methods

### Study area and field sampling

The Yellow River town is located in the northern part of the Zhangqiu district, Jinan city of the Shandong Province, China. The Yellow River has a significant impact on soil and anthropogenic activities because its northwest side is adjacent to the Yellow River. We chose the Yellow River town for this study, and confirmed the research area (36°57′45″-36°59′17″N, 117°15′21″108°31′59″-117°18′10″E) (Fig 1). Curtilage is the land that removed villages and relocated the residences. Curtilage land soil (CS), cropland soil, grassland soil, curtilage-converted cropland soil (CCS) and curtilage-converted grassland soil (CGS) were collected from Yellow River town. Among these, CGS and CGS were the soils that had been converted from rural curtilage for 5 years. The area experiences a temperate monsoon climate. The average annual temperature and precipitation are 12.8°C and 600.8 mm, respectively. The frost-free period is 192 days.

In May of 2021, samples were collected using a soil auger. At each site, a single mixed soil sample was collected using the five-point sampling method within a 0.5 m x 0.5 m quadrat, and the location's coordinates and altitude were recorded using a global positioning system (GPS) device. Four soil samples were collected for each land type. Each soil sample was divided into three parts and stored at –20°C for microbial composition analysis, 4°C for analysis of microbial biomass carbon and nitrogen (MBC & MBN) and dissolved organic carbon and nitrogen (DOC & DON), and ~20°C for LFOM and HFOM analysis. A total of 20 surface soil samples (0–10 cm) were collected. The Fig 1 depicts a detailed distribution of sampling locations.

### Analysis of DOC and MBC

The chloroform fumigation method was conducted on all the soil samples to kill the microbes [21]. A vacuum pump was used to boil 20 mL of chloroform in the drying cabinet after adding 10 g of fresh soil and the sodium hydrate. After vacuuming out the gasified chloroform, the drying cabinet was left overnight to eliminate any lingering microorganisms. Meanwhile, a weighed portion of fresh soil and sodium hydrate solution was placed inside a drying cabinet as control. Both the fumigated and control soils were mixed with 2-M potassium chloride (KCl) solution at 1:5 ratio. After oscillation for 1 h at speed of 200 r/min, the mixtures were centrifugation at 4500 r/min for 10 min and filtration with filter membrane of 0.045 μm. The dissolvable organic matter in supernatant of the control soil was identified as soil DOM, and that of the fumigated soil was the sum of microbial biomass organic matter and soil dissolved organic matter. The C and N contents of the supernatant were determined using a C/N

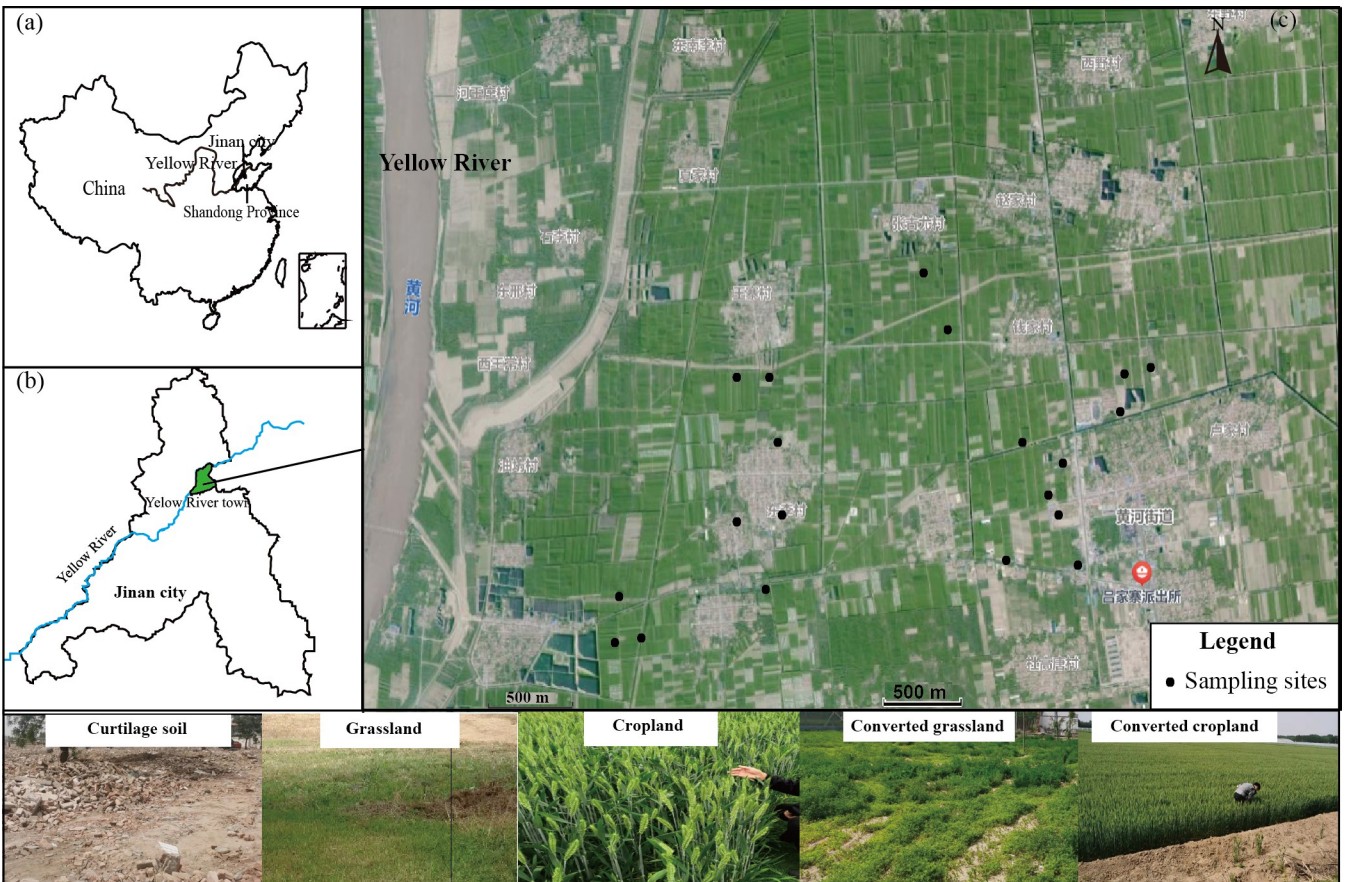

**Fig 1. Sampling sites and surrounding environments.** The site of Jinan city and Shandong Province (a) The site of Yellow River town (b).The sampling sites and surroundings in the Yellow River town (c). The data information is obtained from Landset 8 OLI_TIRS with data identification of LC81220342021147LGN00, which is provided by Geospatial Data Cloud site, Computer Network Information Center, Chinese Academy of Sciences. (http://www.gscloud.cn).

analyzer (ANALYTIKJENA MULTI N/C 3100, Elementar Analysensysteme, Germany), and C and N contents in supernatant of the control soil were the DOC and DON respectively. The MBC and MBN contents were the remainders of C and N in fumigated soil subtracted with those in controlled soil respectively.

In addition, 10 g of fresh soil was extracted using a 2-M of KCl solution in a 1:5 ratio. The organic carbon and organic nitrogen contents of the leaching solution were analyzed, and corresponding DOC and DON contents were calculated [15]. Moreover, soil pH was determined by a pH meter.

## LFOM and HFOM analysis

Samples were air-dried, ground, and sieved through a 0.9 mm sieve. Subsequently, 10 g of each soil sample was weighed and combined with 40 mL of a 1.85-gmL$^{-1}$ sodium iodide solution for the soil stratification [24]. The light fraction (LF) was afloat and the heavy fraction (HF) was deposited [25]. The 10-min of vltrasonic oscillation and 10-min centrifugation were conducted with the mixtures for absolutely stratification. Then the LF was filtered through 0.043-mm brass sieve. The procedure was repeated 3 times to ensure complete separation. Then, by adding the 0.1 M the calcium chloride solution instead of sodium iodide solution, the procedure

was repeated 5–6 times until all I⁻ reactions ceased. The diluted hydrochloric acid (1:10 of volume) was added to remove soil inorganic carbon and repeated the above step. Moreover, this step was repeated once or twice by adding ultrapure water until all Cl⁻ reactions ceased. Finally, the LF and HF were transferred into weighed beakers and oven-dried at 40°C. LF and HF were weighed as LFOM and HFOM respectively. Both the elements of C and N in LFOM and HFOM were determined by using an element analyzer (Vario EL III, Elementar Analysensysteme, Germany), and the contents of LFOC, LFON, HFOC and HFON were calculated [10].

## DNA extraction and Illumina sequencing

The soil was freeze-dried and sieved, and DNA from each weighted soil sample was extracted using the DNA isolation kit (MO-BIO Laboratories, Carlsbad, CA, USA) following the manufacturer's guidelines. Then, the DNA was identified and quantified using 2% agarose gel electrophoresis and Nanodrop ND-1000 UV-Vis spectrophotometer (Thermo Fisher Scientific, Waltham, MA, USA), respectively [26]. Moreover, the DNA was diluted to 1 ng/μL for further analysis. Polymerase chain reaction (PCR) was conducted to amplify the V3-V4 region using the primer (F: ACTCCTACGGGAGGCAGCA and R: TCGGACTACHVGGGTWTCTAAT) and Phusion High-Fidelity PCR Master Mix with GC Buffer and efficient high-fidelity enzymes to maintain accuracy. Furthermore, 2% agarose gel electrophoresis and the gel recovery kit (Qiagen Gel Extraction Kit, Germany) were used to examine and recover the PCR products [27]. The DNA sequencing library was produced, quantified, detected, and paired-end sequenced using the NovaSeq6000 (Illumina, San Diego, CA, USA) platform [28].

The paired-end reads were excised by Qiime2 cutadapt trim-paired process, and the sequence of the unmatched primer was discarded. The paired-end reads were merged to obtain the raw tags. Qiime quality control process was used to filter the raw tags and then the high-quality sequence was obtained. Chimeric sequences were detected and removed by using Qiime dada2 denoise-paired process. Then, UPARSE (www.drive5.com/uparse/) was used to cluster the high-quality sequences into operational taxonomic units with 97% similarity threshold. Finally, the operational taxonomic units were taxonomically annotated against the SILVA132 reference database (http://www.arb-silva.de/). The data of bacterial sequences were uploaded in NCBI with the serial number of PRJNA878090.

## Statistical analysis

The mean analysis and one-way analysis of variance were conducted to analyze the mean values and statistical significance of the C and N fractions across the five soil types. Pearson correlation analysis was used to examine the probable connections between C and N fractions, pH, and bacterial composition. Principal coordinates analysis based on the Bray–Curtis distance was implemented according to bacterial composition [26]. Moreover, bacterial function prediction was analyzed using a Phylogenetic Investigation of Communities by Reconstruction of Unobserved States (PICRUSt2) based on the data library. Chao1 Index and Observed Species Index were calculated to determine bacterial richness, and Shannon Index was proceeded to analyze the bacterial diversity (http://scikit-bio.org/docs/latest/generated/skbio.diversity.alpha.html#module-skbio.diversity.alpha) [28–30]. In addition, the method of neural network analysis was conducted to analyze the microbial significance in relation to OM fractions by SPSS v22.0. Summarily, the SPSS v22.0, QIIME2 (2019.4), and R Studio v4.0 were used to analyze data, and Origin v2018, Adobe Illustrator CS5, and ArcGIS v10.2.2 were used to create figures. P<0.05 was considered statistically significant and "*" indicated statistically significant differences.

# Results and discussion

## The characteristics of SOM fractions

The OM fractions between cropland and grassland showed different contents and composition characteristics. The contents of DOC were sorted as: CS (8.36 mg·kg$^{-1}$) < CCS (11.38 mg·kg$^{-1}$) < CGS (16.49 mg·kg$^{-1}$) < cropland (17.95 mg·kg$^{-1}$) < grassland (18.19 mg·kg$^{-1}$). The results indicated that: 1) CS and the converted soil showed notably low DOC content; 2) DOC contents were higher in grassland than in cropland, and land conversion had no effect (Fig 2). Previous studies indicated high DOC contents in grassland than in cropland [21, 31], Guggenberger et al. [32] suggested that plenty of mineral composition of CS was not prone to the DOC retention. Meanwhile, abundant microbial degradation activities in unfavorable and barren soil environments probably play a decisive role for the accumulation of DOC [4]. Thus, the high DOC content in the converted soil than in CS may indicate the improved microbial degradation, and the remarkable DOC content in CGS than CCS suggested that grassland can effectively guide the DOC accumulation and slow down the carbon mineralization.

Both MBC and MBN contents were reportedly very sensitive to different land use, on the one hand, composition and abundance of microorganism may vary geographically [22, 33], on the other hand, different land use showed distinctive microbial community characteristics due to management practices, human activities, root secretion, etc.[8, 34]. In this study, MBC contents were sorted as: CGS (23.46 mg·kg$^{-1}$) < CCS (40.67 mg·kg$^{-1}$) < CS (55.80 mg·kg$^{-1}$) < grassland (75.73 mg·kg$^{-1}$) < cropland (111.86 mg·kg$^{-1}$; Fig 2). MBC contents increased from CS to grassland and to cropland, suggesting the high and low microbial abundance in cropland

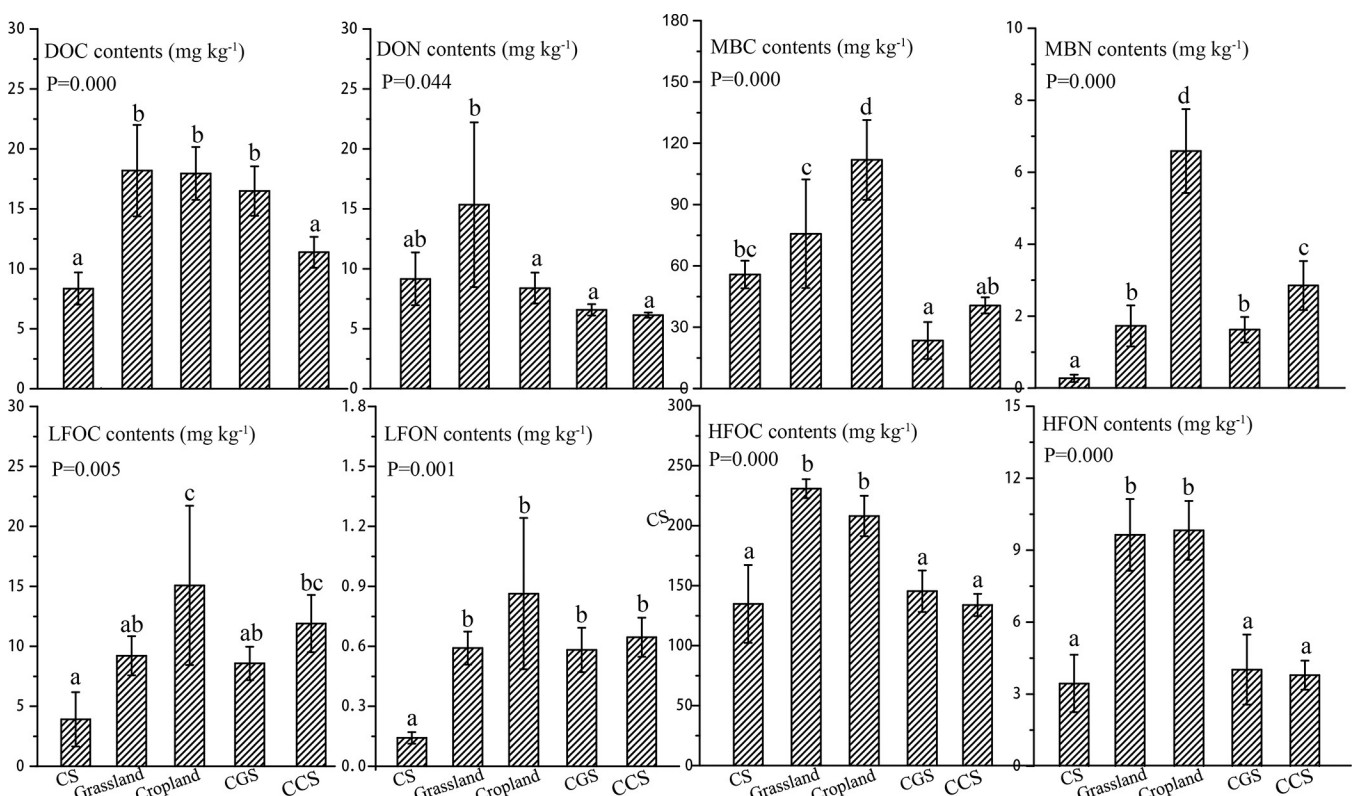

**Fig 2. Contents of C and N fractions in the five land-use types.** Different letters on bars in each carbon fraction indicated significant differences at α = 0.05 level (Duncan test).

**Table 1. Mean values of bacterial richness and diversity indices.**

| Land use types | Chao1 Index | Observed Species Index | Shannon Index |
| --- | --- | --- | --- |
| Curtilage land | 5233a[1] | 4335a | 10.6a |
| Grassland | 6711b | 5631b | 10.9ab |
| Cropland | 8749c | 6914c | 11.6c |
| CGS | 7450b | 5933b | 11.3bc |
| CCS | 6610b | 5209ab | 10.8ab |

[1] The same letters in each column indicated non-significant difference at α = 0.05 level (Duncan test).

and CS, respectively, which were also indicated by the Chao1 Index and Observed Species Index (Table 1). It suggested that regular crop rotation, fertilization and irrigation were prone to the continuous supply of OM and inorganic nutrients, which resulted in a high microbial abundance and diversity [35]. Contrarily, constructed land tended to had low bacterial richness and diversity, it was coincident with previous studies [36]. For example, Quadros et al. [37] found that both bacterial diversity and biomass were low in mined soil and urban soil.

Interestingly, our results suggested that CGS and CCS had remarkably high bacterial abundance and diversity while low MBC contents in contrast with CS. Smith et al. [38] showed that mesotrophic grassland tended to have high abundance of fungal communities and fungal bacterial ratio. Reversely, the significantly low content of dissolved organic nitrogen and high nutritional requirement of CGS and CCS may reduce the fungal abundance. In addition, the Chao1 Index and Observed Species Index were notably higher in cropland than in CCS while those values were the reverse in grassland and CGS. Previous report indicated the low bacterial richness and diversity in transformed grassland than agricultural grassland [26, 39]. The finding indicated the reverse effect of land conversion on cropland and grassland. Variations in bacterial composition might have affected the results (Table 1).

Microbes could degrade, absorb, and transfer the LFOM within soil [10]. LFOC and LFON presented similar distribution patterns, which were significantly the lowest (3.91 and 0.14 mg·kg$^{-1}$ respectively) in CS and the highest (15.08 and 0.86 mg·kg$^{-1}$ respectively) in cropland. Plant residue returning to the field served as a significant source of LFOM accumulation [20], and it presented an insignificant difference due to land conversion. HFOM dominated either the C or N pool in soil due to its stability [16]. HFOC and HFON showed significantly higher accumulation in grassland and cropland than in curtilage soil, suggesting that grassland and cropland environment facilitated the formation and storage of stable organic matter. While curtilage land conversion had a unobvious effect on the HFOM content. Straw retention and plant cover were reported can promote stable organic carbon accumulation [16], while it was inapplicable when faced with curtilage land conversion. Wang et al. [40] demonstrated the reverse distribution pattern of MBC and HFOC among various land types, microbial richness and composition were likely important factors affecting the stable C accumulation.

## The composition characteristic of bacterial communities

The dominant soil bacteria belonged to the phyla of Proteobacteria, Actinobacteria, Acidobacteria, Chloroflexi, Firmicutes, Gemmatimonadetes, and Bacteroidetes (Fig 3). The composition and abundance of microorganism differed considerably among the studied land types. The proportions of Proteobacteria and Acidobacteria in cropland were 35.18% and 18.13%, respectively, significantly higher than in other soil types. CGS (31.13%) and CCS (31.85%) had significantly higher Actinobacteria abundance than grassland (26.58%) and cropland (20.91%), whereas CS had the lowest Actinobacteria abundance (14.71%). Based on the degradation

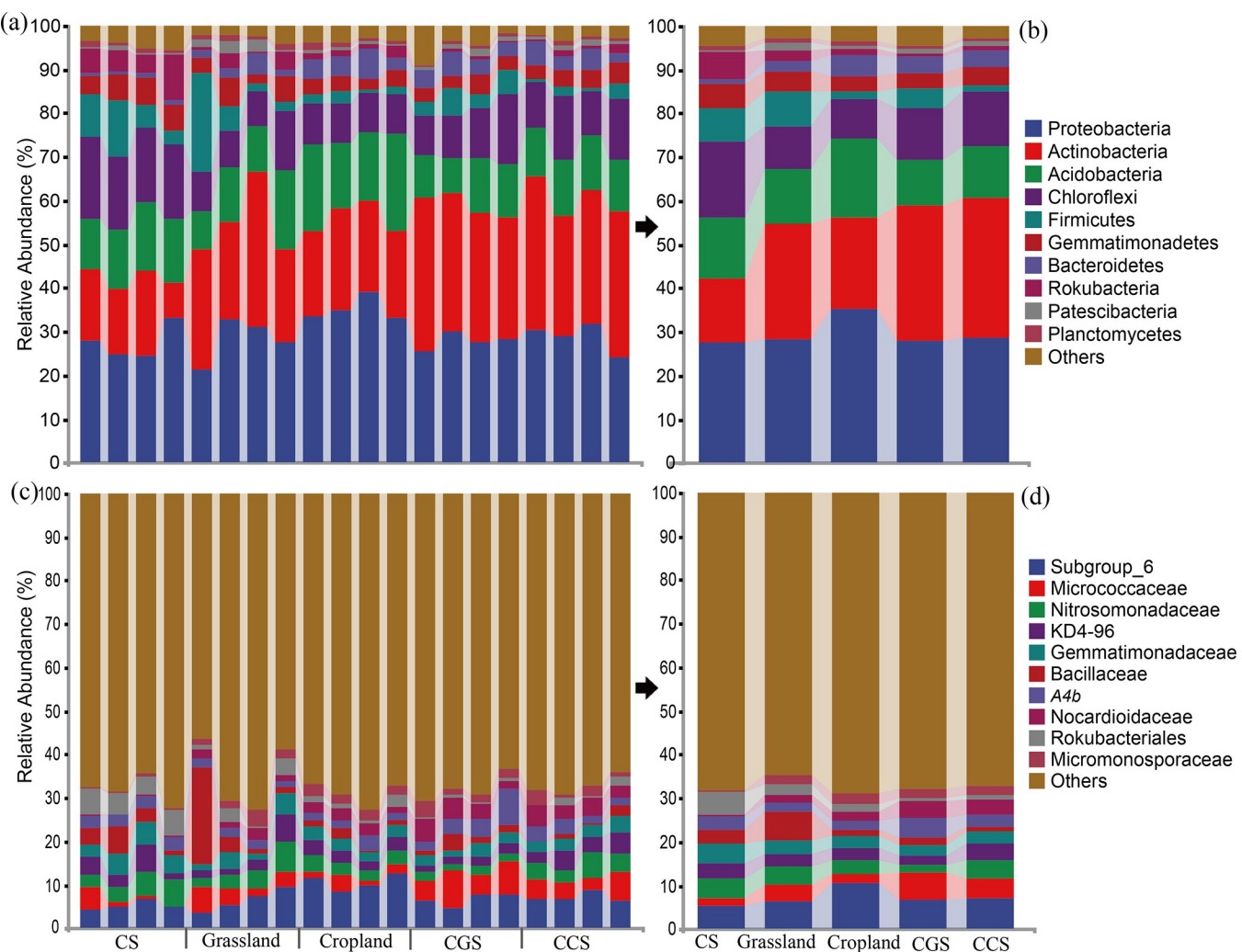

**Fig 3. Taxonomic composition analysis in levels of phyla and families in the five soil types.** Bacterial composition in phyla level of all the samples (a), phylum level of the five soil types (b), family level of all the samples (c), family level of the five soil types (d).

function of long-chain n-alkanes and other organics of Actinobacteria [41], CS and the converted soil presented low and high OM degradation capacity respectively in this study. At the family level, Subgroup 6, Micrococcaceae, Nitrosomonadaceae, KD4-96, and Gemmatimonadaceae were particularly common (Fig 3). The prevalence of Subgroup 6 (10.66%) in cropland may be attributed to the high contents of OC fractions and the relatively neutral soil pH [42]. A high proportion of Micrococcaceae in CGS, CCS, and grassland soils is indicative of high carbon mineralization based on its degradation capabilities [43]. Previous study showed sensitive bacterial composition among land use types, proportions of *Bacillus*, *Pseudomonas* and *Rhodococcus* were the most prominent, which were also significantly different in this study (Fig 3) [44].

The principal coordinates analysis of all the soil sampling sites revealed a cluster of CGS and CCS samples, a cluster of CS samples, and a cluster of cropland samples (S1 Fig). This result demonstrated that land conversion has a greater effect on microbial composition than land-use types. Roadside soil, as the most prolific area for grass growth in the countryside, is susceptible to anthropogenic interference. The scattered distribution of the four grassland

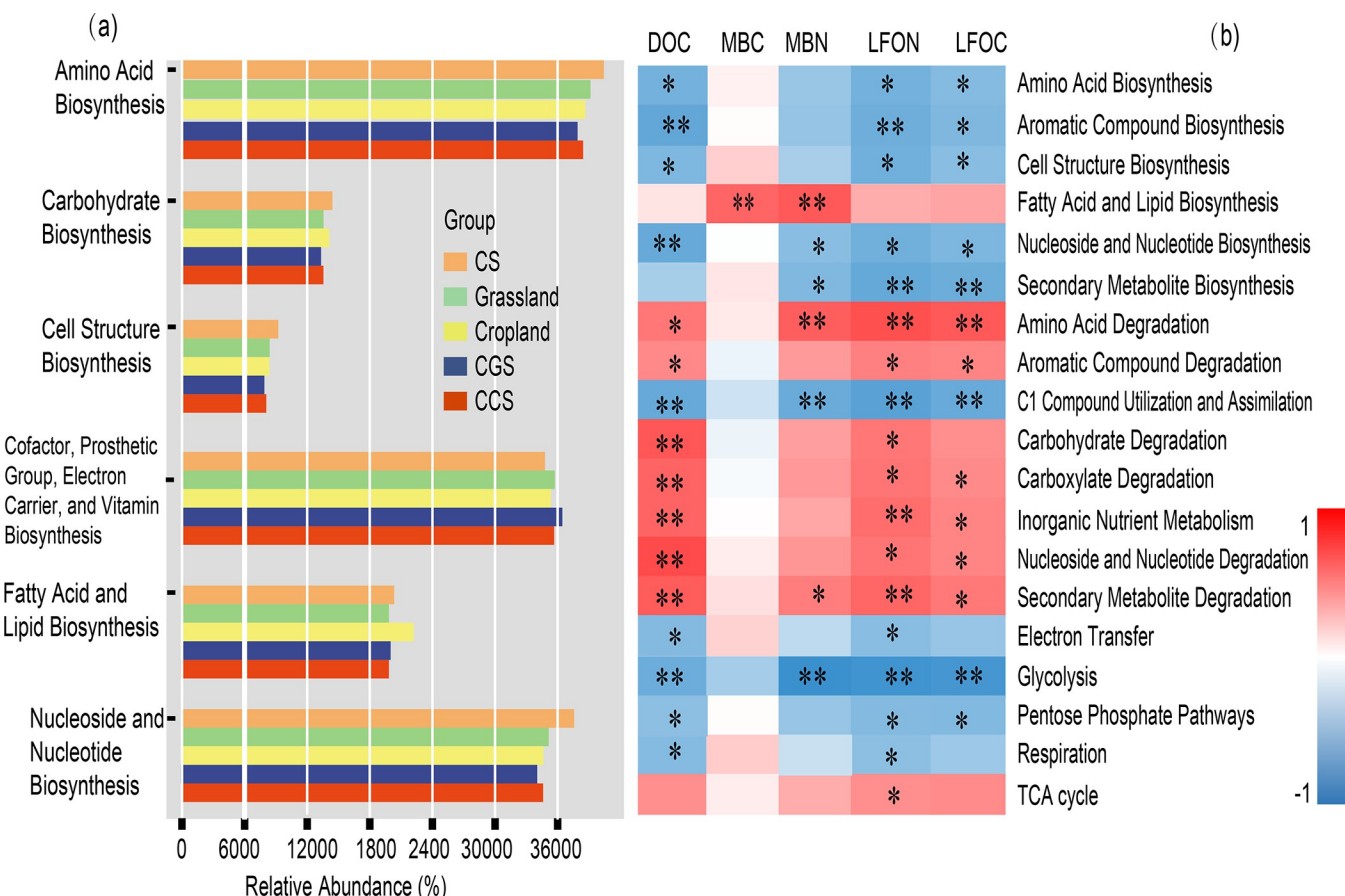

**Fig 4.** Bacterial relative abundance of main biosynthesis function (a) and the Pearson correlation analysis between OM fractions and major microbial functional genes (b). The "**" meant significance was less than 0.01, and "*" meant significance was less than 0.05.in the five soil types.

samples showed notable variations in microbial composition, which can probably be attributed to anthropogenic activities.

PICRUSt2 analysis identified the abundance of bacterial functional genes, which including the biosynthesis, degradation, detoxification, assimilation, etc. The most abundant microbial functional genes were involved in the biosynthesis of nucleoside and nucleotide, amino acid, carbohydrate, and cell structure, and the generation of precursor metabolite and energy (Fig 4), whereas those involved in degradation, utilization, and assimilation were relatively rare (S1 Table). The high functional gene abundance of biosynthesis in CS may indicate that soil microbes proliferated after being released from the perennial high pressure. With an average relative abundance of 4179, the cytochrome c biosynthesis in aerobic respiration was the dominant bacterial activity [45]. The Fig 5 depicted the bacterial composition and contribution to cytochrome c biosynthesis. It showed that Proteobacteria, Actinobacteria, Acidobacteria and Chloroflexi were the main phyla, and Subgroup 6, Nitrosomonadaceae, KD4-96, Gemmatimonadacea, and Bacillaceae were the dominant families. The high abundance of cytochrome c biosynthesis in CS further suggested the rapid expansion of aerobic microorganism abundance (Fig 5).

## Relationship between OM fractions and bacterial communities

The microbial significance in relation to OM fractions was analyzed using a neural network, with OM fractions and microbial phyla proportions serving as dependent variables and

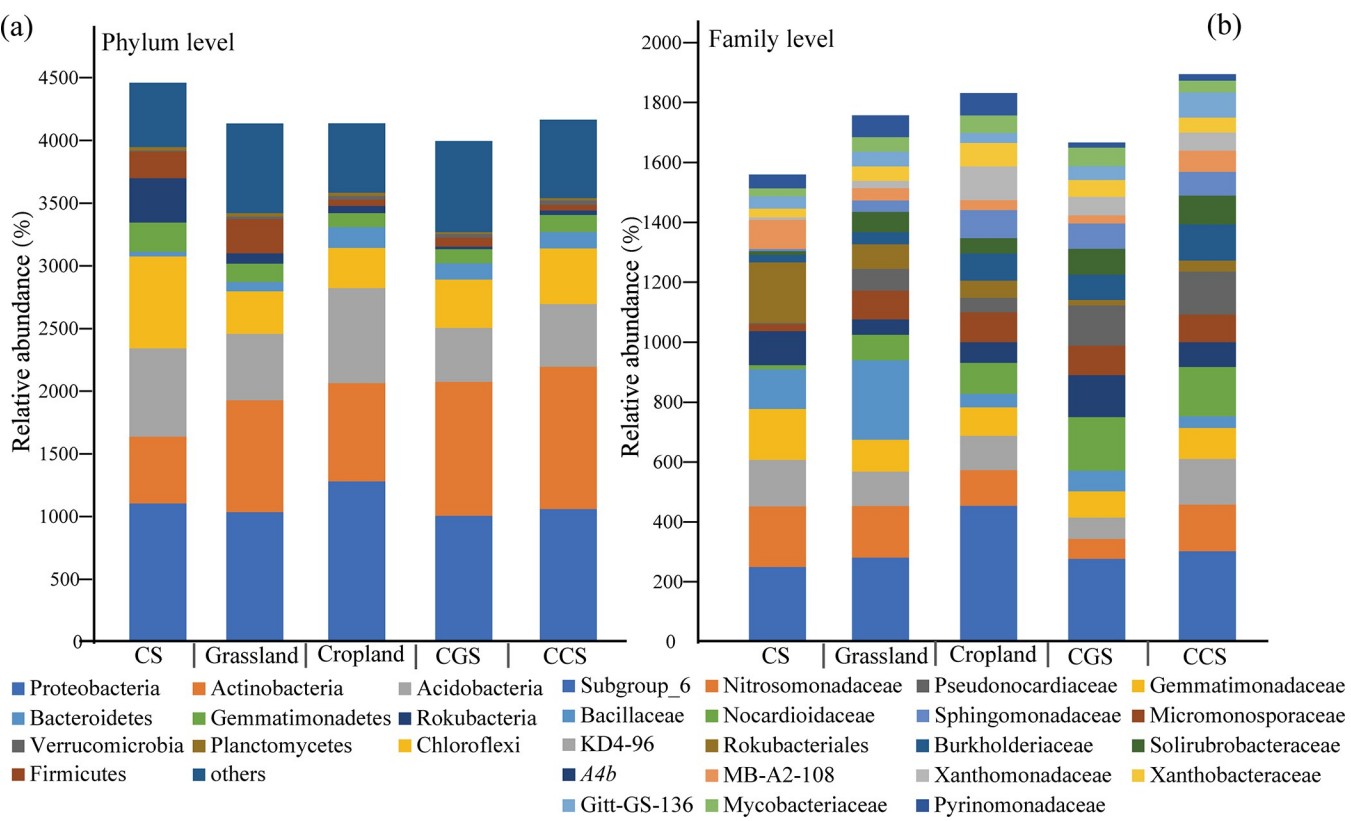

**Fig 5. Bacterial relative abundance of cytochrome c biosynthesis function in aerobic respiration based on the MetaCyc (https://metacyc.org/) metabolic pathway prediction.**

covariants, respectively (S2 Fig). Results revealed that Planctomycetes contributed to the accumulation of DOC and HFOC significantly. Previous report indicated that soil management history and environmental heterogeneity had a substantial effect on Planctomycetes and, in turn, the HFOC content [16, 46]. Planctomycetes also had notable associations with allochthonous and autochthonous tyrosine-like DOM components [46]. Based on the results of neural network, we hypothesized that Planctomycetes were involved in the production and transformation of both DOM and HFOM. Therefore, the dominance of Planctomycetes was directly responsible for the high contents of DOC, DON, HFOC, and HFON in grassland. Moreover, Acidobacteria had the greatest influence on MBC and MBN. Studies by Chaves et al. [47] and Xu et al. [48] demonstrated the capability of Acidobacteria to degrade complex OM, such as hemicellulose, in low-nutrient environments. Thus, Acidobacteria promoted the OM transformation and accumulation of MBC and MBN by degrading biological residues. Thus, the notably low Acidobacteria abundance in the CGS and CCS contributed to the low MBC and MBN contents. LFOC and LFON provided nutrition to heterotrophic microorganisms; thus, they presented positive and significant associations with bacterial richness and diversity indexes.

Actinobacteria abundance was significantly and directly correlated with OC degradation [43]. In this study, Actinobacteria was positively associated with the DOC proportion (DOC: SOC; $R^2 = 0.404$, $p = 0.003$) and negatively related with MBC proportion (MBC: SOC; $R^2 = 0.255$, $p = 0.023$). The findings suggested that Actinobacteria, particularly in CCS and CGS, promoted MBC degradation and DOC production. Micrococcaceae is a common family of Actinobacteria that can resist adverse soil environments due to its degradation capability [49];

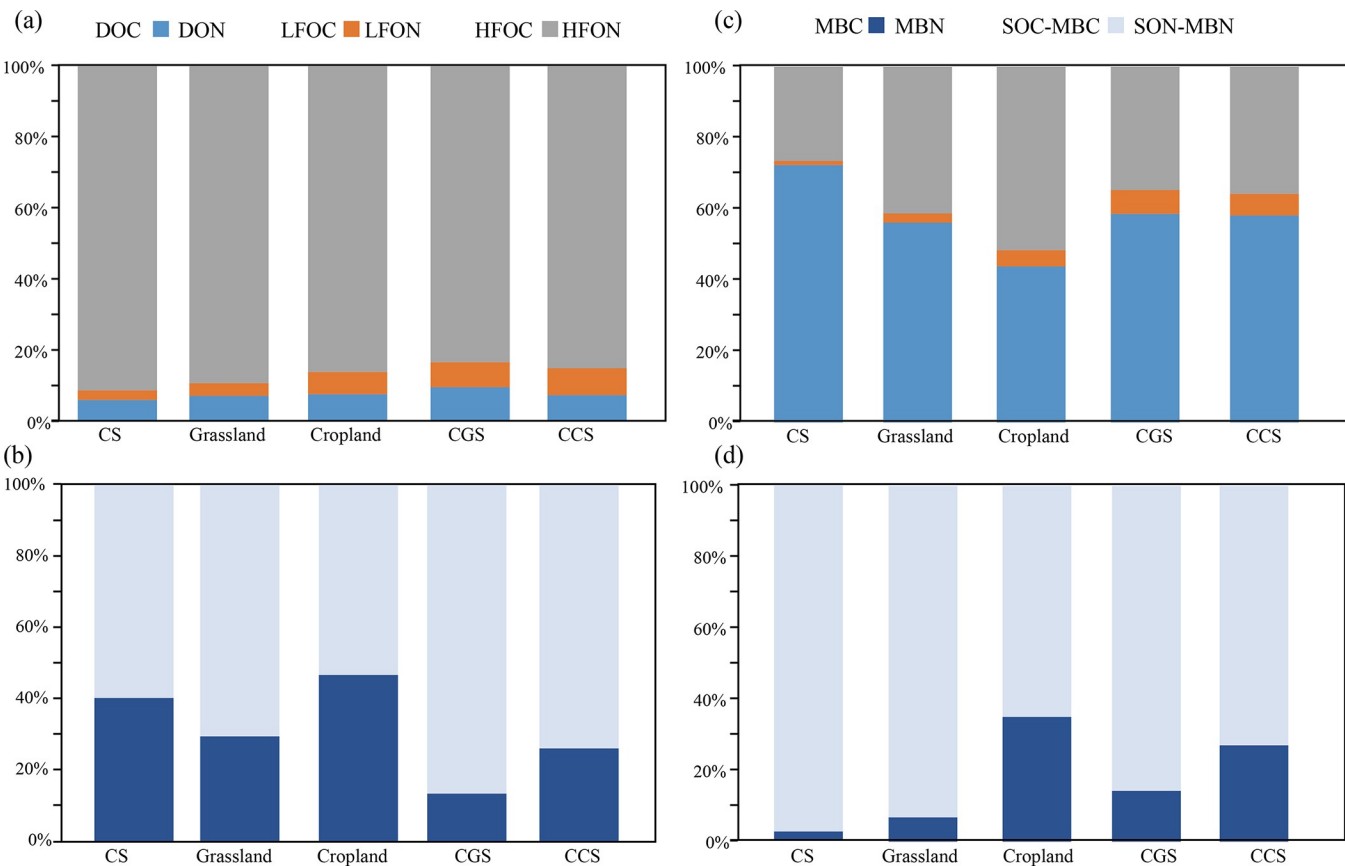

**Fig 6. Proportions of C fractions and N fractions.** The (a) showed proportions of DOC, LFOC, HFOC; (b) showed proportions of MBC and other organic carbon; (c) showed proportions of DON, LFON, HFON; (d) showed proportions of MBN and other organic nitrogen.

its marked association with MBC content indicated the substantial depletion of microbial-derived C. Therefore, microorganisms in the converted soil were in a "hungry" state. Gamboa et al. [50] reported that land use change from regeneration to reforestation significantly reduced MBC content and microbial activities. Zhang et al. [51] also indicated the significant decrease of MBC and bacterial abundance when soil was changed from wildwood to cropland or plantations. Thus MBC and microorganism were remarkably sensitive to land change. Meanwhile, malign microbial composition and nutritional metabolism aggravated the above-mentioned effect in the curtilage converted soil. However, in cropland, the relative abundance of functional genes in the fatty acid and lipid biosynthesis was notably high (Fig 5), and it was positively correlated with the content of MBC and MBN (Fig 4B). The findings suggested that cropland microorganism was in a "fat" state with affluent C and N fractions. Tang et al. [52] showed that cultivation-induced soil nutrient loss could enhance microbial depletion, and Yuan et al. [53] reported that grassland and plantation had higher MBC contents than cropland. Therefore, the findings may indicate that regular nutrient inputs and land management were important measures for soil MBC and microbe stabilization. DOC, LFOC and MBC—all of which represent active OC—had a significant positive correlation, and they were also significantly related to Chao1, Observed Species, and Shannon indices (S2 Table). Thus, they provided an abundance of nutrients and energy for microbial development and metabolism. Both LFOC and DOC, as the major undecomposed or partly decomposed products of complex SOM, can be significantly affected by MBC content and microbial composition [28].

The pH, the most important soil physicochemical parameter, can either directly or indirectly influence the microbial community by affecting the composition of SOM [53]. After curtilage conversion, pH presented a decreasing trend (S3 Fig), but it remained alkaline with values as follows: CS (9.04) > CCS (8.91) > CGS (8.59) > grassland (8.28) > cropland (8.17). The previous study showed that high pH and alkalescent soil habitats were prone to accumulate OM [54], while the extremely significant and negative relationships between the OM fractions and pH suggested the adverse effect of alkaline soil on OM storage, microbial richness and diversity (S2 Table). In contrast, plant roots can exude substantial amounts of organic acids [26], which contributed to the decrease in pH values from CS to converted soil. In response to changes in the environment, the diversity of soil bacteria can also change. In general, soil pH had minimal effects on the relative abundance of bacterial functional genes (S4 Fig).

HFOC and HFON also had no notable association with the main functional genes, suggesting that they are resistant to bacterial interference. Moreover, DOC, LFOC, and LFON were significantly and negatively correlated with most functional genes involved in the biosynthesis and generation of precursor, metabolite, and energy (such as electron transfer). The findings might suggest that contents of the C and N fractions can regulate bacterial functional activities [49]. Positive correlations between DOC, LFOC, LFON, and bacterial degradation activities provided additional evidence that they had numerous microbial degraded products.

## Carbon accumulation potential

The soil organic carbon (SOC) was the sum of DOC, LFOC and HFOC contents based on its separation steps, thus the proportion of each C fraction was determined by comparing it with the SOC, so was the proportion of each N fraction (Fig 6). For land-use types, grassland had the highest SOC content (258.39 mg·kg$^{-1}$) than cropland and CS, among which, MBC proportion of grassland was significantly low. It may indicate the low efficiency of microbial metabolism, which facilitated the physicochemical wrapping and protection of OC particles, resulting in the OC accumulation and storage (Fig 6A and 6B). Ding et al. [55] also reported the advantage of microbial residue C accumulation in grassland. While CS showed the low proportions of DOC and LFOC, as well as the lowest contents of SOC. It illustrated small input limited the OC accumulation. For land conversion, CGS and CCS showed notably higher proportions of DOC and LFOC while lower MBC than CS, which led to markedly higher SOC accumulation in the converted soils. The findings also showed that low proportion of MBC was prone to C storage. Rafeza et al. [56] reported low emission of carbon dioxide with relatively low contents of MBC, which also confirmed the negative effect of MBC on OC accumulation. In contrast to grassland and cropland, CGS and CCS showed high LFOC, similar DOC while low MBC proportion. It suggested relatively low metabolism efficiency of LFOC in the converted soils.

The proportions of N fractions were also calculated and analyzed (Fig 6C and 6D). In this study, DON and HFON were the dominant fractions with proportion ranged of 43.87~72.37% and 36.48~51.52% respectively. For land types, soil organic nitrogen (SON) content was sorted as: grassland (25.57 mg·kg$^{-1}$) > cropland (19.08 mg·kg$^{-1}$) > CS (12.74 mg·kg$^{-1}$) notably. Among that, CS showed significantly higher DON and lower HFON proportions, indicating the low stability. The cropland had extremely higher proportion of MBN (35.07%), which may benefit from increased microbial biomass, enzymatic activities, and ON utilization and consumption [57].

The C: N ratio is a good indicator of C and N mineralization in soil [58]. The notably high values of C: N in CGS (14.86) and CCS (15.34) in comparison to the low value of grassland (10.10) indicated a high rate of C and N mineralization. Theoretically, MBC: LFOC and MBC: DOC values could represent the quantity of unit MBC that degraded the total LFOC or produced the total DOC, respectively (S5 Fig). both the two values had significant differences

among the five land-use types, which were also inter-related by an exponential function with an $R^2$ of 0.736. Further analysis suggested that both LFOC consumption and DOC production were negatively regulated by Actinobacteria.

## Conclusions

With the process of rural relocation and combination, the soil quality of curtilage-converted land should be focused. This is the first study to describe SOM and bacterial community composition characteristics in rural curtilage soil and potential changes when converted to cropland and grassland. We find that the given bacterial composition promotes the degradation of microbial-derived C, resulting in a microbial "hungry" state and notably low MBC content in curtilage-converted grassland and cropland. Reversely, microorganism is in a "fat" state based on the significant abundance of biosynthesis related genes, high values of MBC content, bacterial richness and diversity. Therefore, effectively improving the soil MBC contents is the key to optimize soil quality. This study improves the understanding and efficient use of curtilage converted land. Several detailed measures should be conducted to activate and fatten the bacteria in curtilage soil.

## Supporting information

**S1 Fig. The principal coordinates analysis based on the Bray-Curtis distance.**
(TIF)

**S2 Fig. The importance of microbial phyla to the OM fractions.**
(TIF)

**S3 Fig. The distribution of pH in constructive land soils (CS), grassland soils (GS), wheat soils (WS), converted grassland soils (CGS), and converted wheat soils (CWS).** Different letters on bars indicated significant differences among the soil types analyzed by Duncan test.
(TIF)

**S4 Fig. The Pearson correlation analysis among organic matter fractions, pH, and bacterial composition.**
(TIF)

**S5 Fig. The carbon and nitrogen ratio (C: N) of dissolved organic carbon and nitrogen (DOC and DON), microbial biomass carbon and nitrogen (MBC and MBN), light fraction organic carbon and nitrogen (LFOC: LFON), and heavy fraction organic carbon and nitrogen (HFOC: HFON).** Different letters on bars indicated significant differences among the soil types analyzed by Duncan test.
(TIF)

**S1 Table. The relative abundance of dominant functional genes in the five soil types.**
(DOC)

**S2 Table. The Pearson correlation analysis among organic matter fractions, pH, and bacterial richness and diversity indexes.**
(DOC)

## Acknowledgments

We thank Shanghai Personal Biotechnology Co., Ltd. (Shanghai, China) for microbial Illumina Sequencing. We also thank Dr. Junyu Dong for the revision. The authors thank Editage Company for improving the quality of English of this manuscript.

## Author Contributions

**Conceptualization:** Qingqing Cao, Wen Ma.

**Data curation:** Qingqing Cao, Jinhang Wu, Wen Ma, Dongxu Cui.

**Formal analysis:** Bing Liu.

**Funding acquisition:** Qingqing Cao, Dongxu Cui.

**Investigation:** Bing Liu, Jinhang Wu.

**Methodology:** Qingqing Cao, Wen Ma.

**Project administration:** Dongxu Cui.

**Resources:** Wen Ma.

**Software:** Jinhang Wu, Xu Zhang.

**Supervision:** Bing Liu, Dongxu Cui.

**Validation:** Bing Liu, Xu Zhang.

**Writing – original draft:** Qingqing Cao, Dongxu Cui.

**Writing – review & editing:** Qingqing Cao, Bing Liu, Jinhang Wu, Xu Zhang, Wen Ma.

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
