## [Decision Letter · Decision Letter 0]

27 Jul 2022

PONE-D-22-15913SOC fractions accumulation and microbial characteristic in curtilage soil: effects of land conversion as well as land usePLOS ONE

Dear Dr. Cui,

Thank you for submitting your manuscript to PLOS ONE. After careful consideration, we feel that it has merit but does not fully meet PLOS ONE’s publication criteria as it currently stands. Therefore, we invite you to submit a revised version of the manuscript that addresses the points raised during the review process.

We look forward to receiving your revised manuscript.

Kind regards,

Upendra Kumar, Ph.D.

Academic Editor

PLOS ONE

Journal Requirements:

   "Yes. This study is supported by the National Key Research and Development Plan “Research on the spatial optimization and layout of rural and communities” (NO. 2019YFD1100801), recieved by Dongxu Cui. Prof Cui participates in the study design, data analysis and manuscript writing."

    "We thank Shanghai Personal Biotechnology Co., Ltd. (Shanghai, China) for microbial Illumina Sequencing. We also thank Dr. Junyu Dong for the revision. This study was supported by the National Key Research and Development Plan “Research on the spatial optimization and layout of rural and communities” (NO. 

2019YFD1100801)."

   "Yes. This study is supported by the National Key Research and Development Plan “Research on the spatial optimization and layout of rural and communities” (NO. 2019YFD1100801), recieved by Dongxu Cui. Prof Cui participates in the study design, data analysis and manuscript writing." 

6. Thank you for stating the following in your Competing Interests section:  

    "NO authors have competing interests"

7. We note that you have stated that you will provide repository information for your data at acceptance. Should your manuscript be accepted for publication, we will hold it until you provide the relevant accession numbers or DOIs necessary to access your data. If you wish to make changes to your Data Availability statement, please describe these changes in your cover letter and we will update your Data Availability statement to reflect the information you provide.

8. PLOS requires an ORCID iD for the corresponding author in Editorial Manager on papers submitted after December 6th, 2016. Please ensure that you have an ORCID iD and that it is validated in Editorial Manager. To do this, go to ‘Update my Information’ (in the upper left-hand corner of the main menu), and click on the Fetch/Validate link next to the ORCID field. This will take you to the ORCID site and allow you to create a new iD or authenticate a pre-existing iD in Editorial Manager. Please see the following video for instructions on linking an ORCID iD to your Editorial Manager account: https://www.youtube.com/watch?v=_xcclfuvtxQ

9. We note that Figure 1 in your submission contain [map/satellite] images which may be copyrighted. All PLOS content is published under the Creative Commons Attribution License (CC BY 4.0), which means that the manuscript, images, and Supporting Information files will be freely available online, and any third party is permitted to access, download, copy, distribute, and use these materials in any way, even commercially, with proper attribution. For these reasons, we cannot publish previously copyrighted maps or satellite images created using proprietary data, such as Google software (Google Maps, Street View, and Earth). For more information, see our copyright guidelines: http://journals.plos.org/plosone/s/licenses-and-copyright.

10. Please include captions for your Supporting Information files at the end of your manuscript, and update any in-text citations to match accordingly. Please see our Supporting Information guidelines for more information: http://journals.plos.org/plosone/s/supporting-information. 

Reviewers' comments:

Reviewer's Responses to Questions

**Comments to the Author**

1. Is the manuscript technically sound, and do the data support the conclusions?

Reviewer #1: Partly

Reviewer #2: Partly

2. Has the statistical analysis been performed appropriately and rigorously? 

Reviewer #1: Yes

Reviewer #2: Yes

3. Have the authors made all data underlying the findings in their manuscript fully available?

Reviewer #1: Yes

Reviewer #2: Yes

4. Is the manuscript presented in an intelligible fashion and written in standard English?

Reviewer #1: No

Reviewer #2: No

5. Review Comments to the Author

Reviewer #1: Dear Authors,

The manuscript require rigorous effort to improve its quality of presentation. Specifically, suggestions have been given in the reviewed manuscript itself. In general, the suggestions have been mentioned below:

1. Abstract: Rewrite abstract. Do not repeat the sentences, quantify your finding to understand how LUC has improved the OM fractions (1-2 lines). Also, first mention about results of SOC fraction followed by microbial high-throughput analysis. Use full and abbreviations appropriately.

2. Introduce your topic accurately. Do not use irreverent information that have no scientific background. Use appropriate unit for presentation. Several statements have not been found linked to generated proper hypothesis. Go through attached reviewed manuscript for more clarification.

3. Methodology should be clear. Use full form of any words along with its abbreviation at first time. Write about indices and its equation. How it was determined? No such information is present in the M&M. Use appropriate unit. Explain about new terminology. such as curtilage soil, tidal soil.

4. Result and discussion: Not written well. Many information present in R&D have not been mentioned in M&M.

5. Improve the resolution of figures.

6. Conclusion: Rewrite it and concise the conclusion.

Regards,

Reviewer #2: 1. Abstract should be more focused with a line of research gap, hypothesis and few best quantitative data.

2. Very recent relevant citation must be incorporate in introduction and discussion section.

3. Line no. 137: Rewrite the “℃”,

4. Line no. 173: Only write 10g instead of “10.00 g”

5. Materials and methods: Write a bit details of DNA sequencing steps you have followed like trimming, no of reads generated, contig, aligning etc.

6. Result and discussion should follow as per materials and methods (M&M) sequence. Therefore, resequence your M&M like-wise; otherwise start discussion with the result of LFOM and HFOM.

7. Line no. 264-265: Reason for italicized the family name “family level, Subgroup 6, Micrococcaceae, Nitrosomonadaceae, KD4-96, and Gemmatimonadaceae”??

8. Elaborately discuss the section “Bacterial composition in different soil types” with recent reports and try to establish a strong correlation with perspective treatments.

9. In Fig 1: Direction map should layer in this figure.

10. Conclusion should be modified as per the best result and a line of recommendation.

11. Spacing issue should be carefully checked, besides, please revise whole MS once again for at least fluent scientific English language.

6. PLOS authors have the option to publish the peer review history of their article (what does this mean?). If published, this will include your full peer review and any attached files.

Reviewer #1: **Yes: **Rajeev Padbhushan

Reviewer #2: **Yes: **Megha Kaviraj

---

## [Author Response · Author response to Decision Letter 0]

8 Sep 2022

Response to the editor and reviewers’ suggestions point-by-point 

Responses to Editor's specific comments:

We have revised and rechecked the manuscript format based on the files you given. Thanks very much! 

2.We suggest you thoroughly copyedit your manuscript for language usage, spelling, and grammar. If you do not know anyone who can help you do this, you may wish to consider employing a professional scientific editing service. 

Thanks for you suggestions, language usage of this manuscript is revised by Editage Company (www.editage.com) that you recommended. Thanks for your coupon.

Thanks for your comments. We removed the funding information in “acknowledgement” part.

   "Yes. This study is supported by the National Key Research and Development Plan “Research on the spatial optimization and layout of rural and communities” (NO. 2019YFD1100801), recieved by Dongxu Cui. Prof Cui participates in the study design, data analysis and manuscript writing."

Thanks for your comments. The relevant research of this manuscript is supported by The National key research and development program “Research on the spatial optimization and layout of rural and communities” (NO. 2019YFD1100801). Prof. Dongxu Cui is the program director. The funders had no role in study design, data collection and analysis, decision to publish, or preparation of the manuscript.

    "We thank Shanghai Personal Biotechnology Co., Ltd. (Shanghai, China) for microbial Illumina Sequencing. We also thank Dr. Junyu Dong for the revision. This study was supported by the National Key Research and Development Plan “Research on the spatial optimization and layout of rural and communities” (NO. 

2019YFD1100801)."

   "Yes. This study is supported by the National Key Research and Development Plan “Research on the spatial optimization and layout of rural and communities” (NO. 2019YFD1100801), recieved by Dongxu Cui. Prof Cui participates in the study design, data analysis and manuscript writing." 

Thanks for your comment.We removed the funding information in “acknowledgement” part. This study is supported by the National Key Research and Development Plan “Research on the spatial optimization and layout of rural and communities” (NO. 2019YFD1100801). Prof. Dongxu Cui is the program director. The funders had no role in study design, data collection and analysis, decision to publish, or preparation of the manuscript.

6. Thank you for stating the following in your Competing Interests section:  

    "NO authors have competing interests"

Thanks for your comment. "The authors have declared that no competing interests exist." is stated in the revised cover letter.

7.We note that you have stated that you will provide repository information for your data at acceptance. Should your manuscript be accepted for publication, we will hold it until you provide the relevant accession numbers or DOIs necessary to access your data. If you wish to make changes to your Data Availability statement, please describe these changes in your cover letter and we will update your Data Availability statement to reflect the information you provide.

Thanks for your comment. We can update the microbial data to NCBI when submit the revised manuscript. This has stated in the revised cover letter.

8.PLOS requires an ORCID iD for the corresponding author in Editorial Manager on papers submitted after December 6th, 2016. Please ensure that you have an ORCID iD and that it is validated in Editorial Manager. To do this, go to ‘Update my Information’ (in the upper left-hand corner of the main menu), and click on the Fetch/Validate link next to the ORCID field. This will take you to the ORCID site and allow you to create a new iD or authenticate a pre-existing iD in Editorial Manager. Please see the following video for instructions on linking an ORCID iD to your Editorial Manager account: https://www.youtube.com/watch?v=_xcclfuvtxQ

Thanks for your comment, Prof. Dongxu Cui has registered the ORCID iD, which is 0000-0003-1962-1459.

9.We note that Figure 1 in your submission contain [map/satellite] images which may be copyrighted. All PLOS content is published under the Creative Commons Attribution License (CC BY 4.0), which means that the manuscript, images, and Supporting Information files will be freely available online, and any third party is permitted to access, download, copy, distribute, and use these materials in any way, even commercially, with proper attribution. For these reasons, we cannot publish previously copyrighted maps or satellite images created using proprietary data, such as Google software (Google Maps, Street View, and Earth). For more information, see our copyright guidelines: http://journals.plos.org/plosone/s/licenses-and-copyright.

10. 

11.We require you to either (a) present written permission from the copyright holder to publish these figures specifically under the CC BY 4.0 license, or (b) remove the figures from your submission:

12. 

13.a. You may seek permission from the original copyright holder of Figure 1 to publish the content specifically under the CC BY 4.0 license.  

14. 

15.We recommend that you contact the original copyright holder with the Content Permission Form (http://journals.plos.org/plosone/s/file?id=7c09/content-permission-form.pdf) and the following text:

16.“I request permission for the open-access journal PLOS ONE to publish XXX under the Creative Commons Attribution License (CCAL) CC BY 4.0 (http://creativecommons.org/licenses/by/4.0/). Please be aware that this license allows unrestricted use and distribution, even commercially, by third parties. Please reply and provide explicit written permission to publish XXX under a CC BY license and complete the attached form.”

17. 

18.Please upload the completed Content Permission Form or other proof of granted permissions as an "Other" file with your submission.

19. 

20.In the figure caption of the copyrighted figure, please include the following text: “Reprinted from [ref] under a CC BY license, with permission from [name of publisher], original copyright [original copyright year].”

21. 

22.b. If you are unable to obtain permission from the original copyright holder to publish these figures under the CC BY 4.0 license or if the copyright holder’s requirements are incompatible with the CC BY 4.0 license, please either i) remove the figure or ii) supply a replacement figure that complies with the CC BY 4.0 license. Please check copyright information on all replacement figures and update the figure caption with source information. If applicable, please specify in the figure caption text when a figure is similar but not identical to the original image and is therefore for illustrative purposes only.

23.The following resources for replacing copyrighted map figures may be helpful:

24. 

25.USGS National Map Viewer (public domain): http://viewer.nationalmap.gov/viewer/

26.The Gateway to Astronaut Photography of Earth (public domain): http://eol.jsc.nasa.gov/sseop/clickmap/

27.Maps at the CIA (public domain): https://www.cia.gov/library/publications/the-world-factbook/index.html and https://www.cia.gov/library/publications/cia-maps-publications/index.html

28.NASA Earth Observatory (public domain): http://earthobservatory.nasa.gov/

29.Landsat: http://landsat.visibleearth.nasa.gov/

30.USGS EROS (Earth Resources Observatory and Science (EROS) Center) (public domain): http://eros.usgs.gov/#

31.Natural Earth (public domain): http://www.naturalearthdata.com/

Thanks for your suggestions. We drew the Figure 1 by referring the Geospatial Data Cloud site, Computer Network Information Center, Chinese Academy of Sciences. (http://www.gscloud.cn). In the revised manuscript, we revised the Figure 1 and noted the referring information. 

32.Please include captions for your Supporting Information files at the end of your manuscript, and update any in-text citations to match accordingly. Please see our Supporting Information guidelines for more information: http://journals.plos.org/plosone/s/supporting-information. 

33.

34.

Thanks for your comment. Figure captions were inserted immediately after the first paragraph in which the figure is cited. We added the Supporting Information in the revised manuscript, and figure and table captions in supplementary file were listed.

Response to the editor and reviewers’ suggestions point-by-point 

Responses to reviewers’ specific comments:

Reviewer #1: Dear Authors,

The manuscript require rigorous effort to improve its quality of presentation. Specifically, suggestions have been given in the reviewed manuscript itself. 

Answer: We greatly appreciate the constructive and detailed comments. The English language of the whole manuscript was revised and checked by Editage Company and all the authors. All your suggestions were adopted and the manuscript was revised based on your suggestions. We hope that our revision and answers will enable our manuscript to win your satisfaction. And we would be happy to do whatever is required if further revision is necessary.

In general, the suggestions have been mentioned below:

1. Abstract: Rewrite abstract. Do not repeat the sentences, quantify your finding to understand how LUC has improved the OM fractions (1-2 lines). Also, first mention about results of SOC fraction followed by microbial high-throughput analysis. Use full and abbreviations appropriately.

Answer: Thanks for your comment. We thoroughly revised the “Abstract” part and reduced the usage of abbreviations. The revised Abstract was showed below.

Abctract:

Conversion of curtilage into cropland or grassland can have substantial effects on soil nutrition and microbial activities; however, these effects remain ambiguous. This is the first study to compare the soil organic carbon (SOC) fractions and bacterial communities in rural curtilage, converted cropland, and grassland compared with cropland and grassland. This study determined the light fraction (LF) and heavy fraction (HF) of organic carbon (OC), dissolved organic carbon (DOC), microbial biomass carbon (MBC), and the microbial community structure by conducting a high-throughput analysis. Curtilage soil had significantly lower OC content, the DOC, MBC, LFOC and HFOC of grassland and cropland soils were 104.11%, 55.58%, 264.17%, and 51.04% higher than curtilage soil averagely. Cropland showed notably high bacterial richness and diversity, with Proteobacteria (35.18%), Actinobacteria (31.48%), and Chloroflexi (17.39%) predominating in cropland soil, grassland soil, and curtilage soil, respectively. Moreover, DOC and LFOC contents of converted cropland and grassland soils were 47.17% and 148.65% higher than curtilage soil while MBC content was 46.24% lower than curtilage soil averagely. Land conversion showed more significant effects on microbial composition than land-use differences. The abundant Actinobacteria and Micrococcaceae population and the low MBC contents indicated a “hungry” microbial state in the converted soil, whereas the high MBC content, Acidobacteria proportion, and relative abundance of functional genes in the fatty acid and lipid biosynthesis indicated a “fat” microbial state in cropland. This study contributes to the improvement of soil fertility and the comprehension and efficient use of curtilage soil.

2. Introduce your topic accurately. Do not use irreverent information that have no scientific background. Use appropriate unit for presentation. Several statements have not been found linked to generated proper hypothesis. Go through attached reviewed manuscript for more clarification.

Answer: Thanks for your comment. For the Introduction part, the sentence “Rural revitalization strategy is firstly reported on October 18th, 2017, it aims to improve the natural, social and economic levels in rural areas” were deleted, the unit was revised. After careful revision, several sentences were revised and polished, the detailed changes were showed in the revised manuscript.

3. Methodology should be clear. Use full form of any words along with its abbreviation at first time. Write about indices and its equation. How it was determined? No such information is present in the M&M. Use appropriate unit. Explain about new terminology. such as curtilage soil, tidal soil.

Answer: Thanks for you comment. We explained the curtilage soil, converted cropland soil and converted grassland soil in the revised manuscript. Meanwhile, we added the full name along with its abbreviation. In addition, we revised the description of experimental procedures and checked the English writing. The detailed changes were showed in the revised manuscript.

4. Result and discussion: Not written well. Many information present in R&D have not been mentioned in M&M.

Answer: Thanks for you comments. We revised this part by polishing the language, replenishing the analysis methods in M&M, improving the writing logic, etc. The detailed changes were showed in the text.

5. Improve the resolution of figures.

Answer: Thanks for your comments, we improved the resolution of figures.

6. Conclusion: Rewrite it and concise the conclusion.

Answer: Thanks for your suggestion. We simplified the Conclusions in the revised manuscript, which showed below.

We investigated the relationships between OM fractions and microbial communities, C and N fractions, microbial composition, and functional gene abundance across five land-use types. Curtilage soil (CS) had the lowest OM fractions but higher LFOM and microbial-OM levels, whereas cropland and grassland soil had comparable HFOM and DOC contents. Cropland exhibited a high bacterial richness and diversity, with Proteobacteria, Actinobacteria, and Chloroflexi predominating in cropland, grassland, and CS, respectively. Cropland and grassland converted from curtilage had higher LFOM and MBN content, the same level of HFOM and DOM, and a significantly lower MBC than CS. Different land types suggested various microbial states based on the bacterial abundance and MBC contents. Therefore, the above results contributed to the poor growth of crops or grass on converted soil but vigorous wheat growth on cropland. This is the first study to describe SOM and bacterial community composition characteristics in rural curtilage soil and potential changes when converted to cropland and grassland. It improved the understanding and efficient use of curtilage converted land. Several measures should be conducted to activate and fatten the bacteria in curtilage soil.

Reviewer #2: 1. Abstract should be more focused with a line of research gap, hypothesis and few best quantitative data.

Answer: Thanks for your suggestions, we modified the text in Abstract and improved its writing logic. Moreover, we added the relevant data. 

2. Very recent relevant citation must be incorporate in introduction and discussion section.

Answer: Thanks for your suggestions, we added several recent published papers in the revised manuscript.

3. Line no. 137: Rewrite the “℃”,

Answer: Thanks for your suggestions, we revised it into “°C”.

4. Line no. 173: Only write 10g instead of “10.00 g”

Answer: Thanks for your suggestions, we revised it as you suggested.

5. Materials and methods: Write a bit details of DNA sequencing steps you have followed like trimming, no of reads generated, contig, aligning etc.

Answer: Thanks for the suggestions, we added the relevant description in the revised manuscript, which also listed below.

The paired-end reads were excised by Qiime2 cutadapt trim-paired process, and the sequence of the unmatched primer was discarded. The paired-end reads were merged to obtain the raw tags. Qiime quality control process was used to filter the raw tags and then the high-quality sequence was obtained. Chimeric sequences were detected and removed by using Qiime dada2 denoise-paired process. Then, UPARSE (www.drive5.com/uparse/) was used to cluster the high-quality sequences into operational taxonomic units with 97% similarity threshold. Finally, the operational taxonomic units were taxonomically annotated against the SILVA132 reference database (http://www.arb-silva.de/).

6. Result and discussion should follow as per materials and methods (M&M) sequence. Therefore, resequence your M&M like-wise; otherwise start discussion with the result of LFOM and HFOM.

Answer: Thanks for your suggestion. As you suggested, we moved the analysis of DOC and MBC ahead in M&M part.

7. Line no. 264-265: Reason for italicized the family name “family level, Subgroup 6, Micrococcaceae, Nitrosomonadaceae, KD4-96, and Gemmatimonadaceae”??

Answer: Thanks for your question, it’s unnecessary to italicized the family name, thus we revised them in the whole manuscript.

8. Elaborately discuss the section “Bacterial composition in different soil types” with recent reports and try to establish a strong correlation with perspective treatments.

Answer: Thanks for your suggestion. We added the relevant papers and discussed the effects of soil-type differences in the revised manuscript.

9. In Fig 1: Direction map should layer in this figure.

Answer: Thanks for your suggestion. We revised the Figure 1 in the revised version.

10. Conclusion should be modified as per the best result and a line of recommendation.

Answer: Thanks for your suggestion. We simplified the Conclusions in the revised manuscript, which showed below.

We investigated the relationships between OM fractions and microbial communities, C and N fractions, microbial composition, and functional gene abundance across five land-use types. Curtilage soil (CS) had the lowest OM fractions but higher LFOM and microbial-OM levels, whereas cropland and grassland soil had comparable HFOM and DOC contents. Cropland exhibited a high bacterial richness and diversity, with Proteobacteria, Actinobacteria, and Chloroflexi predominating in cropland, grassland, and CS, respectively. Cropland and grassland converted from curtilage had higher LFOM and MBN content, the same level of HFOM and DOM, and a significantly lower MBC than CS. Different land types suggested various microbial states based on the bacterial abundance and MBC contents. Therefore, the above results contributed to the poor growth of crops or grass on converted soil but vigorous wheat growth on cropland. This is the first study to describe SOM and bacterial community composition characteristics in rural curtilage soil and potential changes when converted to cropland and grassland. It improved the understanding and efficient use of curtilage converted land. Several measures should be conducted to activate and fatten the bacteria in curtilage soil.

11. Spacing issue should be carefully checked, besides, please revise whole MS once again for at least fluent scientific English language.

Answer: We greatly appreciate the constructive and detailed comments. The English language of the whole manuscript was revised and checked by Editage Company and all the authors. All your suggestions were adopted and the manuscript was revised based on your suggestions. We hope that our revision and answers will enable our manuscript to win your satisfaction. And we would be happy to do whatever is required if further revision is necessary.

---

## [Decision Letter · Decision Letter 1]

16 Jan 2023

PONE-D-22-15913R1

Soil organic carbon fraction accumulation and bacterial characteristics in curtilage soil: Effects of land conversion and land use

PLOS ONE

Dear Dr. Cui,

Thank you for submitting your manuscript to PLOS ONE. After careful consideration, we feel that it has merit but does not fully meet PLOS ONE’s publication criteria as it currently stands. Therefore, we invite you to submit a revised version of the manuscript that addresses the points raised during the review process.

We look forward to receiving your revised manuscript.

Kind regards,

Sudeshna Bhattacharjya

Academic Editor

PLOS ONE

Journal Requirements:

Reviewers' comments:

Reviewer's Responses to Questions

**Comments to the Author**

1. If the authors have adequately addressed your comments raised in a previous round of review and you feel that this manuscript is now acceptable for publication, you may indicate that here to bypass the “Comments to the Author” section, enter your conflict of interest statement in the “Confidential to Editor” section, and submit your "Accept" recommendation.

Reviewer #3: (No Response)

2. Is the manuscript technically sound, and do the data support the conclusions?

Reviewer #3: Partly

3. Has the statistical analysis been performed appropriately and rigorously? 

Reviewer #3: Yes

4. Have the authors made all data underlying the findings in their manuscript fully available?

Reviewer #3: Yes

5. Is the manuscript presented in an intelligible fashion and written in standard English?

Reviewer #3: Yes

6. Review Comments to the Author

Reviewer #3: COMMENTS OF THE REFREE (PONE-D-22-15913_R1)

Dear Authors,

The manuscript still requires some of your quality effort to improve its quality of presentation.

The study is good and is of general importance. The introduction and background are not well structured. The experimental data and material is adequate but representation of the data should be more crispy. Statistical analysis of data is perfect and very sound. Result and discussion part should be rewritten and more focused and elaborative with giving explanation like why the abovesaid phyla are more dominant under specific condition, etc. Specifically, suggestions have been given as following:

P1. Introduce your topic accurately. This part should be crisper indicating the research gap and why the study is so much important on global perspective and need to be more focused.

P2. Methodology should be clear. Use full form of any words along with its abbreviation at first time (DOC and DON).

P3. Write about indices and its equation. How it was determined? No such information is present in the M&M. Use appropriate unit with reference.

P4. In line no. 110. (http://www.gscloud.cn): try to specify the area or it should be readily available and easy to locate.

P5. In line no. 135. Write full form of LFOC and HFOC. Mention the standard procedure of their calculation with proper references. How they are separated by using 0.043 mm sieve and give proper reference for that.

P6. How LFOC and HFOC are related with LFOM and HFOM: Mention it. Which fraction is actually measured and how? Give detailed procedure. If only LFOC and HFOC are measured then how it is converted to LFOM and HFOM as they are not synonymous. Similarly for DOM and DOC?

P7. In line no. 209. Table1. The value should be upto 4 digit.

P8. In line no.278: in place of connection, “relationship” may be written.

P9. Result and discussion: Not written well. Many information present in R&D have not been mentioned in M&M

P10. Improve the resolution of figures.

P11. The discussion part was written very superficially and it should be very focused and elaborative and if, it was recorded or found, then try to give a proper explanation that why it was happened.

P12. Conclusion: Conclusion should be modified as per the best result and a line of recommendation. The conclusion part should not be summary of the result of the research work. You should include how the information generated by you would be useful at global perspective.

P13. In line no.374-75: “therefore, …….cropland”: what does it mean, please clarify?

P14. In figure 6: what do you mean by stable OC and ON. Mention its detail in the text and material and methods part as well.

Overall, this study in this stage is not suitable for publication unless authors provide the details mentioned above.

7. PLOS authors have the option to publish the peer review history of their article (what does this mean?). If published, this will include your full peer review and any attached files.

Reviewer #3: **Yes: **SHRILA DAS

---

## [Author Response · Author response to Decision Letter 1]

8 Feb 2023

Response to Reviewers

Reviewer #3: 

Dear Authors,

The manuscript still requires some of your quality effort to improve its quality of presentation.

The study is good and is of general importance. The introduction and background are not well structured. The experimental data and material is adequate but representation of the data should be more crispy. Statistical analysis of data is perfect and very sound. Result and discussion part should be rewritten and more focused and elaborative with giving explanation like why the abovesaid phyla are more dominant under specific condition, etc. Specifically, suggestions have been given as following:

Overall, this study in this stage is not suitable for publication unless authors provide the details mentioned above.

Answer: We greatly appreciate your encouragement and constructive suggestions. We enriched the research background and importance in the Introduction part, revised and improved the representation and wording in the Materials and Methods part, replenished the relevant references of index calculation. In Result and Discussion part, we discussed the relationships of dominant bacteria and land types. We hope that our revision and answers will enable our manuscript to win your satisfaction. 

P1. Introduce your topic accurately. This part should be crisper indicating the research gap and why the study is so much important on global perspective and need to be more focused.

Answer: Thanks for your suggestions. We restated the research background in China and the research importance in global scale. Meanwhile, we added relevant references to support the views. They were showed in the revised manuscript. 

P2. Methodology should be clear. Use full form of any words along with its abbreviation at first time (DOC and DON). 

Answer: Thanks for your suggestions. We revised them in the new revised manuscript.

P3. Write about indices and its equation. How it was determined? No such information is present in the M&M. Use appropriate unit with reference. 

Answer: Thanks for your suggestions. We added the accordance of index calculation. The added part was: “Chao1 Index and Observed Species Index were calculated to determine bacterial richness, and Shannon Index was proceeded to analyze the bacterial diversity (http://scikit-bio.org/docs/latest/generated/skbio.diversity.alpha.html#module-skbio.diversity.alpha) [28-30].”

P4. In line no. 110. (http://www.gscloud.cn): try to specify the area or it should be readily available and easy to locate.

Answer: Thanks for your suggestions. We added the detailed data identification, which was showed: “ The data information is obtained from Landset 8 OLI_TIRS with data identification of LC81220342021147LGN00, which is provided by Geospatial Data Cloud site, Computer Network Information Center, Chinese Academy of Sciences. (http://www.gscloud.cn).”

P5. In line no. 135. Write full form of LFOC and HFOC. Mention the standard procedure of their calculation with proper references. How they are separated by using 0.043 mm sieve and give proper reference for that.

Answer: Thanks for your suggestions. We added the full name of LFOC and HFOC. Meanwhile, we added detailed analyzing steps of the soil stratification, OM fraction separation and fraction determination. Detailed were showed: “Samples were air-dried, ground, and sieved through a 0.9 mm sieve. Subsequently, 10 g of each soil sample was weighed and combined with 40 mL of a 1.85 g·mL-1 sodium iodide solution for the soil stratification [24]. The light fraction (LF) was afloat and the heavy fraction (HF) was deposited [25]. The 10-min of vltrasonic oscillation and 10-min centrifugation were conducted with the mixtures for absolutely stratification. Then the LF was filtered through 0.043-mm brass sieve. The procedure was repeated 3 times to ensure complete separation. Then, by adding the 0.1 M the calcium chloride solution instead of sodium iodide solution, the procedure was repeated 5-6 times until all I- reactions ceased. The diluted hydrochloric acid (1:10 of volume) was added to remove soil inorganic carbon and repeated the above step. Moreover, this step was repeated once or twice by adding ultrapure water until all Cl- reactions ceased. Finally, the LF and HF were transferred into weighed beakers and oven-dried at 40 °C. LF and HF were weighed as LFOM and HFOM respectively. Both the elements of C and N in LFOM and HFOM were determined by using an element analyzer (Vario EL III, Elementar Analysensysteme, Germany), and the contents of LFOC, LFON, HFOC and HFON were calculated [10].”

P6. How LFOC and HFOC are related with LFOM and HFOM: Mention it. Which fraction is actually measured and how? Give detailed procedure. If only LFOC and HFOC are measured then how it is converted to LFOM and HFOM as they are not synonymous. Similarly for DOM and DOC?

Answer: Thanks for your suggestions. We supplied the relations between LFOC, HFOC and LFOM, HFOM, rewrote the detailed procedure. Meanwhile, we also added the detailed analyzing steps of DOC and DON, and explained the relations between DOC, DON and DOM. This part was revised as follows: 

Analysis of DOC and MBC

The chloroform fumigation method was conducted on all the soil samples to kill the microbes [21]. A vacuum pump was used to boil 20 mL of chloroform in the drying cabinet after adding 10 g of fresh soil and the sodium hydrate. After vacuuming out the gasified chloroform, the drying cabinet was left overnight to eliminate any lingering microorganisms. Meanwhile, a weighed portion of fresh soil and sodium hydrate solution was placed inside a drying cabinet as control. Both the fumigated and control soils were mixed with 2-M potassium chloride (KCl) solution at 1:5 ratio. After oscillation for 1 h at speed of 200 r/min, the mixtures were centrifugation at 4500 r/min for 10 min and filtration with filter membrane of 0.045 μm. The dissolvable organic matter in supernatant of the control soil was identified as soil DOM, and that of the fumigated soil was the sum of microbial biomass organic matter and soil dissolved organic matter. The C and N contents of the supernatant were determined using a C/N analyzer (ANALYTIKJENA MULTI N/C 3100, Elementar Analysensysteme, Germany), and C and N contents in supernatant of the control soil were the DOC and DON respectively. The MBC and MBN contents were the remainders of C and N in fumigated soil subtracted with those in controlled soil respectively. 

In addition, 10 g of fresh soil was extracted using a 2-M of KCl solution in a 1:5 ratio. The organic carbon and organic nitrogen contents of the leaching solution were analyzed, and corresponding DOC and DON contents were calculated [15]. Moreover, soil pH was determined by a pH meter. 

LFOM and HFOM analysis

Samples were air-dried, ground, and sieved through a 0.9 mm sieve. Subsequently, 10 g of each soil sample was weighed and combined with 40 mL of a 1.85 g·mL-1 sodium iodide solution for the soil stratification [24]. The light fraction (LF) was afloat and the heavy fraction (HF) was deposited [25]. The 10-min of vltrasonic oscillation and 10-min centrifugation were conducted with the mixtures for absolutely stratification. Then the LF was filtered through 0.043-mm brass sieve. The procedure was repeated 3 times to ensure complete separation. Then, by adding the 0.1 M the calcium chloride solution instead of sodium iodide solution, the procedure was repeated 5-6 times until all I- reactions ceased. The diluted hydrochloric acid (1:10 of volume) was added to remove soil inorganic carbon and repeated the above step. Moreover, this step was repeated once or twice by adding ultrapure water until all Cl- reactions ceased. Finally, the LF and HF were transferred into weighed beakers and oven-dried at 40 °C. LF and HF were weighed as LFOM and HFOM respectively. Both the elements of C and N in LFOM and HFOM were determined by using an element analyzer (Vario EL III, Elementar Analysensysteme, Germany), and the contents of LFOC, LFON, HFOC and HFON were calculated [10]. 

P7. In line no. 209. Table1. The value should be upto 4 digit.

Answer: Thanks for your comments. We revised the Table 1 as you suggested.

P8. In line no.278: in place of connection, “relationship” may be written.

Answer: Thanks for your comments. We revised the Table 1 as you suggested.

P9. Result and discussion: Not written well. Many information present in R&D have not been mentioned in M&M

Answer: Thanks for your comments. We rechecked the correspondence between results of R&D and analyzing steps of M&M, and supplied the relevant procedure. The details were showed in the revised manuscript.

P10. Improve the resolution of figures.

Answer: Thanks for your suggestions. We rechecked and improved all the figure resolution. 

P11. The discussion part was written very superficially and it should be very focused and elaborative and if, it was recorded or found, then try to give a proper explanation that why it was happened.

Answer: Thanks for your suggestions. We added the relevant discussion with several references of previous studies. We supplied relevant proper explanations and our deduction. The detailed were showed in the revised manuscript.

P12. Conclusion: Conclusion should be modified as per the best result and a line of recommendation. The conclusion part should not be summary of the result of the research work. You should include how the information generated by you would be useful at global perspective.

Answer: Thanks for your suggestions.we rewrote the Conclusion part as you suggested. The detailed was showed below.

Conclusions

With the process of rural relocation and combination, the soil quality of curtilage-converted land should be focused. This is the first study to describe SOM and bacterial community composition characteristics in rural curtilage soil and potential changes when converted to cropland and grassland. We find that the given bacterial composition promotes the degradation of microbial-derived C, resulting in a microbial “hungry” state and notably low MBC content in curtilage-converted grassland and cropland. Reversely, microorganism is in a “fat” state based on the significant abundance of biosynthesis related genes, high values of MBC content, bacterial richness and diversity. Therefore, effectively improving the soil MBC contents is the key to optimize soil quality. This study improves the understanding and efficient use of curtilage converted land. Several detailed measures should be conducted to activate and fatten the bacteria in curtilage soil.

P13. In line no.374-75: “therefore, …….cropland”: what does it mean, please clarify?

Answer: Thanks for your comment. We deleted this sentence and rewrote the Conclusion part.

P14. In figure 6: what do you mean by stable OC and ON. Mention its detail in the text and material and methods part as well.

Answer: Thanks for your comments. We are very sorry for the mistakes. We re-calculated the proportions of C and N fraction and redrew the Fig 6. based on the OC classification and C fraction separation steps, we determined the soil organic carbon (SOC) by summing the contents of DOC, LFOC and HFOC up, so was the soil organic nitrogen (SON). It is worth mentioning that proportion of MBC and MBN were the MBC: SOC ratio and MBN: SON ratio respectively. The detailed revision was showed below:

Carbon accumulation potential

The soil organic carbon (SOC) was the sum of DOC, LFOC and HFOC contents based on its separation steps, thus the proportion of each C fraction was determined by comparing it with the SOC, so was the proportion of each N fraction (Fig 6). For land-use types, grassland had the highest SOC content (258.39 mg·kg-1) than cropland and CS, among which, MBC proportion of grassland was significantly low. It may indicate the low efficiency of microbial metabolism, which facilitated the physicochemical wrapping and protection of OC particles, resulting in the OC accumulation and storage (Fig 6a and 6b). Ding et al. [56] also reported the advantage of microbial residue C accumulation in grassland. While CS showed the low proportions of DOC and LFOC, as well as the lowest contents of SOC. It illustrated small input limited the OC accumulation. For land conversion, CGS and CCS showed notably higher proportions of DOC and LFOC while lower MBC than CS, which led to markedly higher SOC accumulation in the converted soils. The findings also showed that low proportion of MBC was prone to C storage. Rafeza et al. [57] reported low emission of carbon dioxide with relatively low contents of MBC, which also confirmed the negative effect of MBC on OC accumulation. In contrast to grassland and cropland, CGS and CCS showed high LFOC, similar DOC while low MBC proportion. It suggested relatively low metabolism efficiency of LFOC in the converted soils.

The proportions of N fractions were also calculated and analyzed (Fig 6c and 6d). In this study, DON and HFON were the dominant fractions with proportion ranged of 43.87~72.37% and 36.48~51.52% respectively. For land types, soil organic nitrogen (SON) content was sorted as: grassland (25.57 mg·kg-1) > cropland (19.08 mg·kg-1) > CS (12.74 mg·kg-1) notably. Among that, CS showed significantly higher DON and lower HFON proportions, indicating the low stability. The cropland had extremely higher proportion of MBN (35.07%), which may benefit from increased microbial biomass, enzymatic activities, and ON utilization and consumption [58].

The C: N ratio is a good indicator of C and N mineralization in soil [59]. The notably high values of C: N in CGS (14.86) and CCS (15.34) in comparison to the low value of grassland (10.10) indicated a high rate of C and N mineralization. Theoretically, MBC: LFOC and MBC: DOC values could represent the quantity of unit MBC that degraded the total LFOC or produced the total DOC, respectively (S5 Fig). both the two values had significant differences among the five land-use types, which were also inter-related by an exponential function with an R2 of 0.736. Further analysis suggested that both LFOC consumption and DOC production were negatively regulated by Actinobacteria.

Fig 6. Proportions of C fractions and N fractions. The (a) showed proportions of DOC, LFOC, HFOC; (b) showed proportions of MBC and other organic carbon; (c) showed proportions of DON, LFON, HFON; (d) showed proportions of MBN and other organic nitrogen.

---

## [Decision Letter · Decision Letter 2]

20 Mar 2023

Soil organic carbon fraction accumulation and bacterial characteristics in curtilage soil: Effects of land conversion and land use

PONE-D-22-15913R2

Dear Dr. Dongxu Cui

We’re pleased to inform you that your manuscript has been judged scientifically suitable for publication and will be formally accepted for publication once it meets all outstanding technical requirements.

Kind regards,

Sudeshna Bhattacharjya, Ph.D

Academic Editor

PLOS ONE

Additional Editor Comments (optional):

Reviewers' comments:

Reviewer's Responses to Questions

**Comments to the Author**

1. If the authors have adequately addressed your comments raised in a previous round of review and you feel that this manuscript is now acceptable for publication, you may indicate that here to bypass the “Comments to the Author” section, enter your conflict of interest statement in the “Confidential to Editor” section, and submit your "Accept" recommendation.

Reviewer #3: All comments have been addressed

2. Is the manuscript technically sound, and do the data support the conclusions?

Reviewer #3: Yes

3. Has the statistical analysis been performed appropriately and rigorously? 

Reviewer #3: Yes

4. Have the authors made all data underlying the findings in their manuscript fully available?

Reviewer #3: Yes

5. Is the manuscript presented in an intelligible fashion and written in standard English?

Reviewer #3: Yes

6. Review Comments to the Author

Reviewer #3: (No Response)

7. PLOS authors have the option to publish the peer review history of their article (what does this mean?). If published, this will include your full peer review and any attached files.

Reviewer #3: No

---

## [Editor Report · Acceptance letter]

29 Mar 2023

PONE-D-22-15913R2 

Soil organic carbon fraction accumulation and bacterial characteristics in curtilage soil: Effects of land conversion and land use 

Dear Dr. Cui:

I'm pleased to inform you that your manuscript has been deemed suitable for publication in PLOS ONE. Congratulations! Your manuscript is now with our production department. 

Kind regards, 

on behalf of

Dr. Sudeshna Bhattacharjya 

Academic Editor

PLOS ONE